# High-Performance Photoresistors Based on Perovskite Thin Film with a High PbI_2_ Doping Level

**DOI:** 10.3390/nano9040505

**Published:** 2019-04-01

**Authors:** Jieni Li, Henan Li, Dong Ding, Zibo Li, Fuming Chen, Ye Wang, Shiwei Liu, Huizhen Yao, Lai Liu, Yumeng Shi

**Affiliations:** 1International Collaborative Laboratory of 2D Materials for Optoelectronics Science and Technology of Ministry of Education, College of Optoelectronic Engineering, Shenzhen University, Shenzhen 518060, China; jnli91@szu.edu.cn (J.L.); ddinjlu@hotmail.com (D.D.); lizibo@szu.edu.cn (Z.L.); shiweidanielliu@gmail.com (S.L.); yaohz@szu.edu.cn (H.Y.); liu229019@163.com (L.L.); yumeng.shi@szu.edu.cn (Y.S.); 2College of Electronic Science and Technology, Shenzhen University, Shenzhen 518060, China; 3School of Physics and Telecommunication Engineering, South China Normal University, Guangzhou 510006, China; fmchen@m.scnu.edu.cn; 4Key Laboratory of Material Physics of Ministry of Education, School of Physics and Engineering, Zhengzhou University, Zhengzhou 450052, China; wa0001ye@e.ntu.edu.sg; 5Engineering Technology Research Center for 2D Material Information Function Devices and Systems of Guangdong Province, College of Optoelectronic Engineering, Shenzhen University, Shenzhen 518060, China

**Keywords:** high PbI_2_ doping content, PC-AFM, photoresistor, grain boundary passivation

## Abstract

We prepared high-performance photoresistors based on CH_3_NH_3_PbI_3_ films with a high PbI_2_ doping level. The role of PbI_2_ in CH_3_NH_3_PbI_3_ perovskite thin film was systematically investigated using scanning electron microscopy, X-ray diffraction, time-resolved photoluminescence spectroscopy, and photoconductive atomic force microscope. Laterally-structured photodetectors have been fabricated based on CH_3_NH_3_PbI_3_ perovskite thin films deposited using precursor solution with various CH_3_NH_3_I:PbI_2_ ratios. Remarkably, the introduction of a suitable amount of PbI_2_ can significantly improve the performance and stability of perovskite-based photoresistors, optoelectronic devices with ultrahigh photo-sensitivity, high current on/off ratio, fast photo response speed, and retarded decay. Specifically, a highest responsivity of 7.8 A/W and a specific detectivity of 2.1 × 10^13^ Jones with a rise time of 0.86 ms and a decay time of 1.5 ms have been achieved. In addition, the local dependence of photocurrent generation in perovskite thin films was revealed by photoconductive atomic force microscopy, which provides direct evidence that the presence of PbI_2_ can effectively passivate the grain boundaries of CH_3_NH_3_PbI_3_ and assist the photocurrent transport more effectively.

## 1. Introduction

Organic–inorganic hybrid perovskites have attracted tremendous research attention since they was firstly investigated as a photoactive layer for solar cells in 2009 [1]. The power conversion efficiencies (PCE) of perovskite-based solar cells have achieved a rapid increase within a few years [2,3,4]. The efficiency of single-junction perovskite solar cells has now increased to over 23% and the efficiency of perovskite-on-silicon tandem solar cells is almost 28%, which is higher than that of amorphous silicon and many other semiconductor-based solar cells [5]. The rise of organic–inorganic hybrid perovskites can be attributed to their outstanding physical properties such as high light-absorption efficiency, high carrier mobility, long electron–hole diffusion length and tunable bandgap [6,7,8]. In addition to solar cells, organic–inorganic hybrid perovskite can also be utilized in many other optoelectronic devices, such as photodetectors [9,10,11,12,13], image sensors [14,15,16], X-ray sensors [17,18,19,20], chemical sensors [21], light-emitting diodes, and lasers [22,23,24,25,26].

Photodetectors are the critical component for converting incident light to electrical signal and are broadly utilized in optical communication, environmental monitoring, medical analysis, and astronomy [27,28]. Compared with inorganic semiconductors (such as ZnO, GaN, and Si), organic–inorganic perovskite-based photodetectors possess many advantages in terms of high photoresponsivity, high sensitivity, and fast response speed, in addition to their simple and low-cost processing requirements for large-scale fabrication [29,30,31,32,33]. 

For a certain perovskite material of AMX_3_ stoichiometry, chemical composition is a crucial factor affecting device performance. CH_3_NH_3_PbI_3_ (MAPbI_3_) is one of the most widely studied photoactive layer materials for solar cells and photodetectors [34,35,36,37]. Previous reports have investigated that a slight excess of PbI_2_ in the MAPbI_3_ film is beneficial for solar cell performance. Kim et al. reported the presence of PbI_2_ in MAPbI_3_ significantly reduces the hysteresis of IV characteristics and ion defect migration and results in a high efficiency of 19.75% [38]. Carmona et al. reported that an excess of PbI_2_ can improve the crystallinity of the MAPbI_3_ film and the electron transfer to the TiO_2_ layer, leading to a high average cell efficiency of 18% [39]. Yang et al. showed that PbI_2_ filling grain boundaries of the perovskite not only reduced the number of trap sites but also generated significant band bending [40]. The doping level of PbI_2_ is critical, which plays an important role in the chemical stability and optoelectronic performance of perovskites. Zhang et al. reported that the PbI_2_ formed at grain boundaries of the perovskite can effectively passivate perovskite films and improve the charge separation efficiency with a longer PL lifetime and enhanced open-circuit voltage [41]. Besides, it has also been reported that PbI_2_ may passivate the TiO_2_ interface and reduce the charge recombination [42]. Similarly, an excessive amount of PbI_2_ can also act as an electron blocking layer and facilitate hole transport between the interface of the perovskite and hole transport layer [43]. However, most of the current studies on the role of PbI_2_ are limited to solar cell applications. With increasing research interest in the photo-detection study of perovskites, there is a demand to systematically investigate the role of PbI_2_ in photodetectors.

In this study, we present a high-performance photoresistor photodetector based on MAPbI_3_ with a high PbI_2_ content. Different from previous reports in which only a slight excess of PbI_2_ was proved to be beneficial, the optimized content of PbI_2_ in our study was much higher. Our results demonstrate that a PbI_2_-rich perovskite precursor (MAI:PbI_2_ = 0.5:1) is beneficial to photodetector performance in terms of photocurrent, responsivity, and photocurrent on/off ratio. Furthermore, the stability of the device can be significantly improved with the incorporation of PbI_2_.

## 2. Materials and Methods

### 2.1. Materials and Preparation of Perovskite Precursor Solution

CH_3_NH_3_I (MAI) and PbI_2_ were purchased from Xi’an Polymer Light Technology Corp. Dimethyl formamide (DMF, 99.9%) was purchased from Aladdin. All chemicals were directly used without further purification. For the perovskite precursor solution preparation, MAI and PbI_2_ with various mole ratios (0.25, 0.5, 0.75, 1.0) were dissolved in DMF and stirred overnight at room temperature. PbI_2_ is fixed at 1 mol/L in all precursor solutions. All prepared solutions were filtered with a polytetrafluoroethylene filter (pore size ~0.2 µm) before film deposition by spin-coating method.

### 2.2. Fabrication of the Devices

Indium tin oxide (ITO) glass with a sheet resistance of ~6 Ω sq^−1^ was used as a conductive substrate for device fabrication. The ITO was laser-etched to form electrode-patterns and subsequently ultrasonicated in acetone, isopropanol, ethanol, and deionized water, and blown dry with N_2_. Then the substrates were treated with air plasma for 20 min. After that, perovskite thin films were spin-coated at 4000 rpm for 25 s and annealed on a hot plate at 100 °C for 30 min. The photoresistors prepared with different precursor solutions (MAI-to-PbI_2_ ratio of 0.25:1, 0.5:1, 0.75:1, and 1:1) were denoted as S1, S2, S3, and S4, respectively. All procedures were conducted in a glove box filled with argon. The active area for all photoresistors is 0.02 cm × 0.7 cm (0.014 cm^2^).

### 2.3. Characterization of Perovskite Thin Films and Device Performance 

The surface and cross-section morphology of the perovskite thin film was examined by scanning electron microscopy (FEG SEM SU-70, Hitachi, Tokyo, Japan). The steady-state photoluminescence (PL) spectra measurements were conducted by the WITec’s Raman microscope alpha300 R using 532 nm as excitation source. The time-resolved photoluminescence (TRPL) spectroscopy was measured by FS5 Fluorescence Spectrometer (Edinburgh Instruments, Edinburgh, England). The X-ray diffraction (XRD) measurements were carried out by a Rigaku D/MAX-2500 diffractometer with Cu K_α_ X-ray source. The photoelectronic performance of the devices was measured in air. A fiber-coupled LED (MCWHF2, Thorlabs) was used to provide white light, and different band pass filters were installed on the LED lamp to achieve monochromatic light. The IV curves under dark and white light conditions were measured by an electrochemical station (CHI 760e, CH Instruments Ins., Shanghai, China) in a dark box. The light intensities of white light and monochromatic light were recorded by TES-132 solar power meter and a Thorlabs power meter (PM100D with a S121C standard photodiode power sensor), respectively, and light intensity was adjusted by regulating the LED lamp’s working current. We set the LED lamp in a pulse mode to achieve pulse illumination, which allowed the recording of the time-dependent photocurrent response (I-t) by the electrochemical station. 

Photo conductive atomic force microscopy (PC-AFM) was conducted using a Bruker Dimension ICON AFM. A Pt-Ir coated silicon probe with a force constant of 3 N/m was used for PC-AFM measurement with peak-force TUNA mode. The light source was a 532 nm laser (Changchun New Industries Optoelectronics Technology Co. Ltd., Changchun, China). The laser was introduced by an inverted optical microscope with an objective lens and vertically shined onto the sample. Topography and current images were achieved simultaneously and compared directly. All of the AFM measurements were conducted in the dark and under 532 nm laser illumination. The NanoScope-Analysis software package was used to analyze the AFM images.

## 3. Results and Discussion

The compositions of the as-synthesized perovskite thin films were characterized by XRD (as shown in Figure 1a). For samples S1 to S3, an additional diffraction peak at 12.56° was observed which originated from the reflection of (001) lattice plane of PbI_2_ [34], while the (001) diffraction peak from PbI_2_ became nearly absent in the sample with a MAI:PbI_2_ ratio of 1:1 (sample S4). Three typical peaks of perovskite located at 14.09°, 28.44°, and 31.85° can be observed, corresponding to the reflections from (110), (220), and (310) lattice plane [9,30,39], respectively. As the MAI:PbI_2_ ratio varied from 0.25:1 to 1:1, typical perovskite peaks got enhanced gradually. The relatively strong intensities were probably caused by the preferred crystal orientation of (110). Figure 1b–e shows the SEM morphology of PbI_2_-rich perovskite thin films (S1–S3) and the pure-phase perovskite (S4). The grain size of the thin films is ranging from ~10 nm to ~100 nm, and becomes larger as the MAI:PbI_2_ ratio varies from 0.25:1 to 1:1.

Figure 2a shows the absorption spectra of all the perovskite films with a thickness of ~300 nm. Similar absorption band-edge positions (~770 nm) were observed for all films. Furthermore, the absorption spectra show an obvious increase with the increasing amount of MAPbI_3_ in the range between 450 nm and 850 nm, which can be ascribed to a higher absorption coefficient of MAPbI_3_ than PbI_2_. Furthermore, the peak shoulder observed at 500 nm in the absorption spectra is attributable to the band-edge absorption of PbI_2_ [38]. Steady-state PL spectra of the thin films are shown in Figure 2b and the inset shows the change of PL emission peak positions with various MAI:PbI_2_ ratios. It can be clearly seen that the peak position shifts towards 776 nm as the MAI:PbI_2_ ratio varies from 0.25:1 to 1:1. It is reported that the interplay between inorganic and organic interactions has an important effect on the lattice strain and the band gap of semiconductors [39]. A head-to-tail configuration of organic cations would cause a reduced lattice strain and larger grain size of perovskite materials. This reduced lattice strain leads to lattice expansion and a red shift in PL emission peak position. Here, the SEM results (Figure 1b–e) show that the grain size of perovskite increased with varying MAI:PbI_2_ ratios (from 0.25:1 to 1:1), and a red shift on PL emission peak position (Figure 2b) is observed for these samples. In a recent work, Jones et al. observed that lattice strain is directly related to enhanced defect concentrations and non-radiative recombination on the micro-scale [44]. Here, the PL intensities are stronger for the PbI_2_-rich perovskites compared with the pure-phase perovskite, which is consistent with previous reports [38,39,45]. The presence of PbI_2_ can passivate the grain boundaries of the perovskite films and this passivation effect on PL intensity is dominant compared with lattice strain. The sample with a MAI:PbI_2_ ratio of 0.5:1 (sample S2) shows the strongest PL emission, which suggests an effective suppression of non-radiative recombination.

To investigate the charge carrier dynamics, TRPL spectroscopy was used to compare the carrier lifetime of the two typical perovskite samples, S2 (MAI:PbI_2_ ratio 0.5:1) and S4 (MAI:PbI_2_ ratio 1:1), as shown in Figure 2c,d. The TRPL curves can be fitted by the bi-exponential function
(1)$$I(t)=A_{1}e^{\left(-t/\tau_{1}\right)}+A_{2}e^{\left(-t/\tau_{2}\right)}$$
where *τ*_1_ and *τ*_2_ are the corresponding fast and slow time constant. *A*_1_ and *A*_2_ are the amplitude of the fast and slow decay component, respectively. We further calculate the contribution from the fast (*A*_1_/(*A*_1_ + *A*_2_)) and slow (*A*_2_/(*A*_1_ + *A*_2_)) process, and all the parameters are listed in Table 1. The sample with MAI:PbI_2_ ratio of 1:1 (sample S2) shows a fast time constant of 4.42 ± 0.23 ns and a slow time constant of 16.3 ± 1.30 ns. For the sample with MAI:PbI_2_ ratio of 1:1 (sample S4), the fast and slow time constants are 1.16 ± 0.02 ns and 13.03 ± 0.21 ns, respectively. The short and long decay times are associated with surface and bulk recombination, respectively [46]. The defect states or shallow trapping levels in the grain boundaries of the perovskite will result in PL emission quenching and therefore behave as non-radiative recombination centers [47]. The observed longer PL lifetime in sample S2 suggests that the grain boundaries of MAPbI_3_ have been effectively passivated by PbI_2_ [41]. The defect states or shallow trapping levels in the grain boundaries of the S2 sample are passivated by PbI_2_ in the film; hence, the S2 sample shows a high PL intensity, which is consistent with the steady-state PL results in Figure 2b.

The spatially resolved photo response study of perovskite thin films by PC-AFM provides reliable and location-dependent electrical analysis of photosensitive materials [48]. The PC-AFM measurements are conducted using a Pt-Ir-coated conductive AFM tip. Bias voltages are applied on the conductive ITO substrate and the Pt-Ir tip is held at ground [49]. For the PC-AFM technique, topography and photocurrent of MAPbI_3_ can be recorded simultaneously. Therefore, it is possible to investigate and compare the charge transport process from different regions under exactly the same experimental conditions. The photocurrent measurement combined with topographic images provides a much more accurate way to investigate local defects (especially at the edges) and spatial fluctuation on local charge distributions. The topographies of the two samples are exhibited in Figure 3a,c. Consistent with observations from SEM images, the PbI_2_-rich perovskite thin film had a smaller grain size and smoother surface with an average roughness of ~4.93 nm, while that of the pure-phase perovskite film is ~10.5 nm. The scanning current images as shown in Figure 3b,d reflect the local optoelectronic and electrical properties at a nanoscale spatial resolution. To directly compare the photocurrent generation in the pure MAPbI_3_ and MAPbI_3_-PbI_2_, we measured the current value change with and without light illumination as shown in Figure 3b,d, which displays the corresponding current mapping in the dark and under 532 nm laser illumination for samples S4 (MAI:PbI_2_ ratio 1:1) and S2 (MAI:PbI_2_ ratio 0.5:1). For the two samples, the current intensities under laser illumination are higher than those in dark. By contrast, the photocurrent mapping for S2 (MAI:PbI_2_ ratio 0.5:1) is larger than that of S4 (MAI:PbI_2_ ratio 1:1), which could be due to the grain boundary passivation effect of PbI_2_. When the grain boundary is passivated by PbI_2_, the defects at grain boundaries can be reduced, resulting in more photogenerated charge carriers transferring along the grain boundaries. Therefore, the photocurrent at the grain boundary becomes further enhanced compared with the center of single-crystalline perovskite grain (as shown in Figure 3d). In addition, the current photo response at 1 V is extracted as shown in Figure 3e, which clearly demonstrates a higher current density in the PbI_2_-rich perovskite thin film.

To demonstrate the effect of the chemical composition on the performance of photodetectors, perovskite-based photodetectors with a lateral device structure of ITO/perovskite/ITO were fabricated. Figure 4a schematically shows the stepwise procedures for device fabrication. The IV curves of the devices measured under the white light with an illumination intensity of 4.5 mW/cm^2^ are shown in Figure 4b. It can be clearly observed that all the devices exhibit a linear IV curve, which suggests good Ohmic contact between the ITO and perovskite thin films. Among all devices, the photodetector fabricated from the precursor solution with a MAI:PbI_2_ ratio of 0.5:1 (sample S2) exhibited the highest photocurrent. The dark IV curves of all devices are compared as shown in Appendix A. The addition of PbI_2_ increased the resistance of the perovskite films and reduced the dark current of the devices. The pure-phase perovskite device showed the highest dark current. The time-dependent photocurrent curves of the devices are shown in Figure 4c. For all the devices, upon photo illumination, the photocurrents increased rapidly. The on/off period was set to 2.5 s and all devices kept a repeatable photocurrent level and response speed after several on/off switching cycles, indicating the excellent reproducibility and stability of the devices. The device fabricated from precursor solution with a MAI:PbI_2_ ratio of 0.5:1 shows the highest photocurrent, which is consistent with the results of the IV measurements.

Responsivity (*R*) is an important parameter to assess the performance of a photodetector. *R* can be calculated using the following equation:
(2)$$R=\frac{I_{p}-I_{d}}{PA}$$
where *I_p_*, *I_d_*, *P*, and *A* are the photocurrent, dark current, incident light intensity, and active area of the device, respectively [49]. The responsivities of all devices at a 5 V bias voltage are summarized in Table 2. The S2 device displays the highest responsivity of 64 mA/W, much higher than the rest of devices. The excellent R value suggests that S2 (MAI:PbI_2_ ratio 0.5:1) possesses a high photon-to-electron/hole conversion efficiency. The electrical measurements indicate that an appropriate excessive amount of PbI_2_ in MAPbI_3_ is beneficial for improving the photoelectrical performance. Response time is another important parameter for a photodetector. The rise edges and decay edges in the time-dependent photo response curve were measured under white light illumination (4.5 mW/cm^2^) at 5 V and enlarged as shown in Appendix A. The rise times (τ_r_, defined as the time for the device increasing to 90% of the photocurrent maximum) and decay times (τ_d_, defined as the time for the device decaying to 10% of the photocurrent) are summarized in Table 2. It can be clearly seen that devices fabricated from perovskite-PbI_2_ thin films show a shorter response time than the device fabricated from pure-phase perovskite (sample S4). The S2 (MAI:PbI_2_ ratio 0.5:1) device exhibited an ultrafast response speed with a rise time of 0.86 ms and a decay time of 1.5 ms, which are outstanding values among the reported lateral-structured photodetectors and even heterostructure devices [11].

The spectral selectivity of the S2 (MAI:PbI_2_ ratio 0.5:1) photoresistor was also studied. Figure 5a shows the spectral photo response of the device in the wavelength range of 400–700 nm under a fixed incident light intensity of 13 μW/cm^2^. The device exhibits a broadband photo response characteristic. A maximum responsivity value of 1.25 A/W under 405 nm illumination is achieved. The external quantum efficiency (EQE) is defined by the equation:
EQE = *hcR_λ_*/*eλ*(3)
where *h* is Plank’s constant, *c* is the velocity of the incident light, *R_λ_* is the responsivity, *e* is the electronic charge, and *λ* is the wavelength of the incident light [50]. The maximum EQE of 400% is achieved under 405 nm illumination. The photoconduction results from the electron–hole pairs excited by the illumination light with photon energy larger than the semiconductor band gap. The 405 nm light with larger energy can excite more electron–hole pairs in perovskite thin films, leading to a higher photocurrent.

Figure 5b shows the log-scale dark and light IV curves of the S2 (MAI:PbI_2_ ratio 0.5:1) photoresistor under white light with illumination intensities varying from 0.14 to 4.46 mW/cm^2^. The photocurrent increased with ascending light intensity. Specific detectivity (*D**) is another important parameter to assess the performance of a photodetector and it can be calculated using Equation (4):
(4)$$D^{*}=\frac{RA^{1/2}}{{\left(2eI_{d}\right)}^{1/2}}$$

Moreover, the responsivities and specific detectivities of the S2 device under different light intensities are calculated and plotted in Figure 5c. It can be seen that both the responsivity and specific detectivity decrease with increasing light intensity. Remarkably, the device shows a highest responsivity value of 7.8 A/W and specific detectivity of 2.1 × 10^13^ Jones under 405 nm light with an intensity of 5 μW/cm^2^, which is higher than the commercial Si-based photodetector (4.2 × 10^12^ Jones) [16]. These excellent photo response performance values are some of the best results compared with previously reported MAPbI_3_-based photodetectors [11,51,52,53,54,55], as shown in Table 3. It is noticeable that the responsivity and specific detectivity decreased rapidly when light intensity was below 0.5 mW/cm^2^, while a further increase in intensity would result in a very slow decrease until saturation is reached. This saturation is observed in many other studies [50] and has been attributed to the filling mechanism of the sensitizing centers of the perovskite thin film at a high light intensity. The saturation value is close to Liang’s report (0.6 mW/cm^2^) [56]. Figure 5d gives the EQE of the S2 device at 5 V bias with different incident light intensity. The EQE reaches the highest value of 2.37 × 10^3^% under 5 μW/cm^2^ 405 nm light illumination. The EQE displays nearly the same trend as responsivity and specific detectivity. In the end, we measured the I-t curves of the pristine S2 device under a strong intensity of 224 mW/cm^2^ to study its stability, as shown in Appendix A. The device shows a good repeatability after several on/off cycles. Then, the I-t curve of the same device was measured again after one month, as shown in Appendix A. A high-level photocurrent remained with a ~15% decay compared with the pristine device. In contrast, the pure perovskite devices all decomposed. This result demonstrates that the incorporation of PbI_2_ in perovskite film is beneficial to improve the stability of the device to some extent. 

## 4. Conclusions

In summary, a high-performance lateral-structured photodetector, photoresistor, based on perovskite film with a high PbI_2_ doping level was successfully prepared by simple spin coating method. The photodetector fabricated from precursor solution with the MAI:PbI_2_ ratio of 0.5:1 shows the best device performance, which is higher than the device based on pure perovskite thin film. The device displays a fast response speed with a rise time of 0.86 ms and a decay time of 1.5 ms. Under 405 nm light illumination, a high responsivity of 7.8 A/W, specific detectivity of 2.1 × 10^13^ Jones, and an EQE of 2.5 × 10^3^% were realized. The analysis of local electrical properties demonstrated that the presence of the excess PbI_2_ in the perovskite film improved its photo-electrical properties. The presence of PbI_2_ can effectively passivate the grain boundaries of MAPbI_3_, resulting in a longer PL lifetime and a higher carriers transport efficiency. Furthermore, the device based on PbI_2_-rich perovskite also showed better stability compared with pure-phase perovskite devices. These results indicate that non-stoichiometric perovskites could be beneficial to device performance and chemical stability, which could also be extended to the exploration of lead-free perovskites for the application in high-performance photodetection.

## Figures and Tables

**Figure 1 nanomaterials-09-00505-f001:**
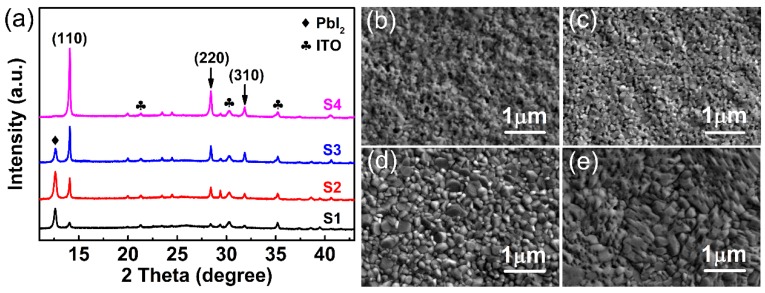
(**a**) XRD patterns of the perovskite films fabricated from precursor solutions with different MAI:PbI_2_ ratios. (**b**) SEM images for the surface morphology of S1 (MAI:PbI_2_ ratio 0.25:1), (**c**) S2 (MAI:PbI_2_ ratio 0.5:1), (**d**) S3 (MAI:PbI_2_ ratio 0.75:1), (**e**) S4 (MAI:PbI_2_ ratio 1:1).

**Figure 2 nanomaterials-09-00505-f002:**
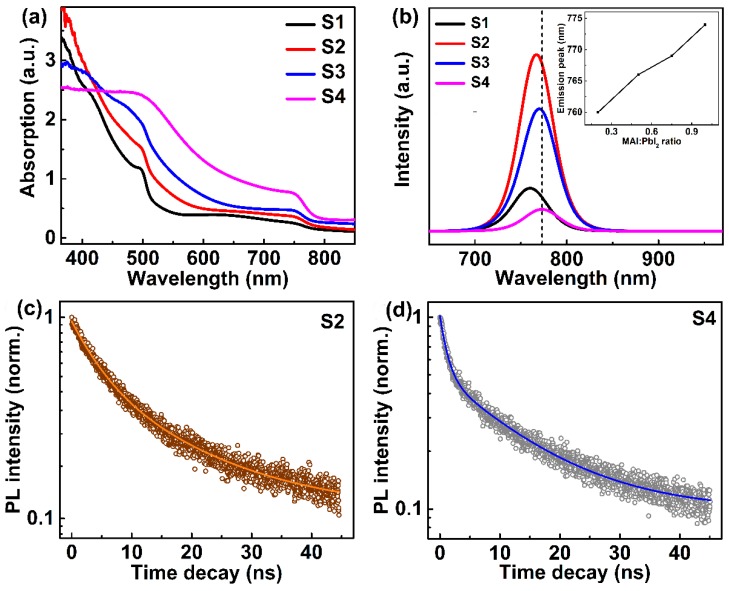
(**a**) Absorption spectra and (**b**) steady-state PL spectra of perovskite thin films with various MAI:PbI_2_ ratio. (**c**) TRPL spectroscopy and corresponding curve fitting of S2 (MAI:PbI_2_ ratio 0.5:1) and (**d**) S4 (MAI:PbI_2_ ratio 1:1).

**Figure 3 nanomaterials-09-00505-f003:**
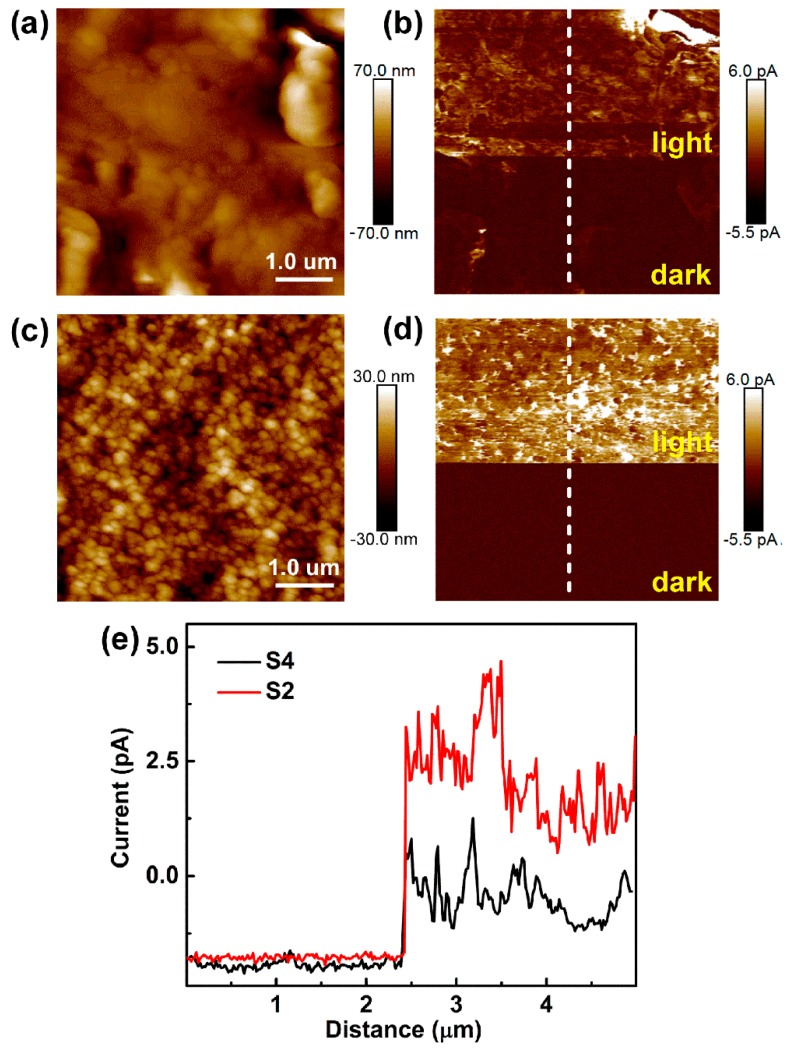
(**a**,**c**) Topographies and (**b**,**d**) photoconductive images of sample S4 (MAI:PbI_2_ ratio 1:1) and S2 (MAI:PbI_2_ ratio 0.5:1) under dark and 532 nm laser illumination, respectively. (**e**) Surface profile of current mapping taken from (**b**) and (**d**) at the location marked by the white dashed lines, showing the surface current change at 1 V of sample S2 (MAI:PbI_2_ ratio 0.5:1) and S4 (MAI:PbI_2_ ratio 1:1) in the dark and under illumination.

**Figure 4 nanomaterials-09-00505-f004:**
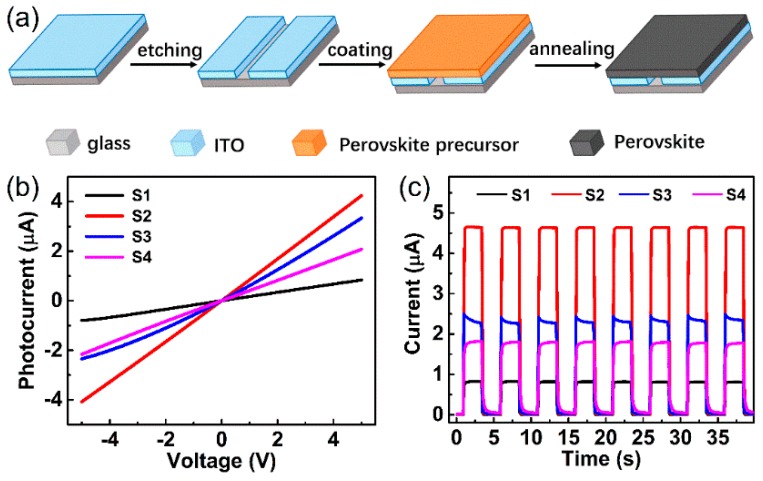
(**a**) Schematic diagram of the stepwise procedures for the fabrication of perovskite-based photodetectors with a device structure of ITO/perovskite/ITO. (**b**) IV measurement taken with a sample bias from −5 V to +5 V. (**c**) Time-dependent photocurrent curves taken at 5 V sample bias. The devices were fabricated based on perovskite films with various MAI:PbI_2_ ratios.

**Figure 5 nanomaterials-09-00505-f005:**
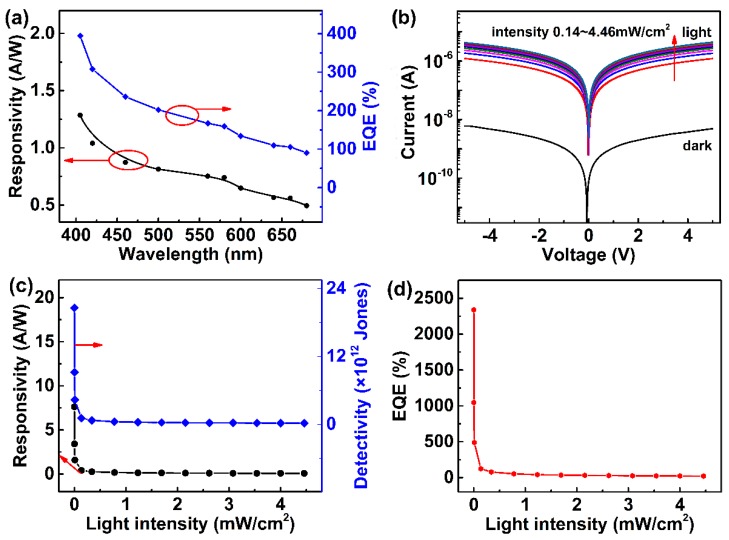
(**a**) Spectral photo response performance of the S2 device at a wavelength ranging from 400 to 700 nm under a fixed incident light intensity of 13 μW/cm^2^. (**b**) The log-scale dark and light IV curves of the S2 photodetector under 405 nm light illumination. (**c**) Responsivity and specific detectivity, (**d**) EQE of the S2 photodetector under 405 nm light with different intensities at a bias of 5 V.

**Table 1 nanomaterials-09-00505-t001:** Bi-exponential fitting result of S2 and S4.

Sample	*A* _1_	*τ*_1_ (ns)	*A* _2_	*τ*_2_ (ns)
S2	0.44 ± 0.03 (52%)	4.42 ± 0.23	0.4 ± 0.03 (48%)	16.3 ± 1.30
S4	0.51 ± 0.01 (55%)	1.16 ± 0.02	0.41 ± 0.01 (45%)	13 ± 0.21

**Table 2 nanomaterials-09-00505-t002:** The summarized parameters of the photodetectors fabricated with various perovskite thin films (sample S1–S4).

Sample	S1	S2	S3	S4
Dark current (uA)	1.787 × 10^−3^	6.018 × 10^−3^	1.254 × 10^−2^	3.961 × 10^−2^
Photocurrent (uA)	0.7993	4.083	2.352	2.161
R (mA/W)	13	64	37	34
on/off ratio	447	678	188	55
t_r_ (ms)	1.25	0.86	2.3	31.9
t_d_ (ms)	1.6	1.5	10.3	385.8

**Table 3 nanomaterials-09-00505-t003:** Photo response performance comparison between this work and other MAPbI_3_-based photodetectors.

Materials	*R* (A/W)	*D** (Jones)	τ_r_/τ_d_ (ms)	Light nm/(mW/cm^2^)	Bias (V)
MAPbI_3_/WS_2_ [11]	17	2 × 10^12^	2.7/7.5	505/0.0002	5
MAPbI_3_ NWs [51]	0.055	5 × 10^10^	150/53	532/40	0.1
MAPbI_3_/TiO_2_ [52]	0.49 × 10^−6^	—	20/20	white/100	3
MAPbI_3_/Rgo [53]	0.074	—	40.9/28.8	520/3.2	5
MAPbI_3_/NAYF_4_:Yb/Er [54]	0.87	5.9 × 10^12^	52/67	730/22.9	2
MAPbI_3_ [55]	8.95	2.9 × 10^12^	7.7/6	532/0.37	10
This work	7.8	2.1 × 10^13^	0.86/1.5	405/0.005	5

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
