# Peer review of "High-Performance Photoresistors Based on Perovskite Thin Film with a High PbI2 Doping Level"

_nanomaterials, 2019, doi:10.3390/nano9040505_

Round 1
Reviewer 1 Report
Manuscript ID: nanomaterials-458223Type of manuscript: ArticleTitle: High-performance photodetectors based on perovskite thin film with ahigh PbI2 doping levelAuthors: Jieni Li, Henan Li *, Dong Ding, Zibo li, Fuming Chen, Ye Wang,Shiwei Liu, Huizhen Yao, Lai Liu, Yumeng Shi
The paper shoes excellent characterization of systematically varied MAPbI3/PbI2 ratio in the precursor solution of the photoactive layers in perovskite photo detectors and should be published. However, the trust in the calculated values of detector properties is damaged due to the following major critic point:
Major critic pont:
Fig. 5a) shows EQE larger than 400% !!!
in line 250 an explanation is given:
The photoconduction results from the electron-hole pairs excited by the illumination light with photon energy larger than the semiconductor band gap. The 405 nm light with larger energy can excite more electron-hole pairs in perovskite thin films, leading to a higher photocurrent.
But 405nm = 3.06 eV and the MAPbI3 energy gap is 1.6 eV à to excite 2 electrons with 1photon at least 3.2 eV would be needed.
Possibly the area of the detector, which is given in line 94 as 0.014 cm2 is way to small it would be 0.12 mm x 0.12 mm.
In addition EQE is a term in photovoltaics; for detectors I find in Wikipedia:
Quantum efficiency of Image Sensors : Quantum efficiency (QE) is the fraction of photon flux that contributes to the photocurrent in a photodetector or a pixel. Quantum efficiency is one of the most important parameters used to evaluate the quality of a detector and is often called the spectral response to reflect its wavelength dependence. It is defined as the number of signal electrons created per incident photon. In some cases it can exceed 100% (i.e. when more than one electron is created per incident photon).
Spectral responsivity
Spectral responsivity is a similar measurement, but it has different units: amperes per watt (A/W); (i.e. how much current comes out of the device per incoming photon of a given energy and wavelength). Both the quantum efficiency and the responsivity are functions of the photons' wavelength (indicated by the subscript λ).
To convert from responsivity (Rλ, in A/W) to QEλ[6] (on a scale 0 to 1):
where λ is the wavelength in nm, h is the Planck constant, c is the speed of light in a vacuum, and e is the elementary charge.
Minor critical points:
1.) luminescence around PbI2 gap i.e. 500nm should be shown?
2.) Red shift with increasing PbI2/MAI ratio related to reduced lattice strain: any hint on strain in XRD?
3.) Line 163: Considering the steady-state PL results as shown in Figure 2b, the observed longer PL lifetime in sample S2 suggests that the grain boundaries of MAPbI 3
But: In Fig.2b) higher luminescence is observed, not lifetime
4.) Line 18
is: In this contribution, we prepared high …
change to: We prepared high …
5.) line 28
is: the special dependence of photocurrent generation
change to: the spacial dependence of photocurrent generation
or change to: the local dependence of photocurrent generation
6.) line 52
is: For a certain perovskite, composition is a crucial factor…
change to: For a certain perovskite material of AMX3 stoichiometry, chemical composition is a crucial factor…
7.) 102
is: photoelectronic performance of the devices were measured in the air
change to: photoelectronic performance of the devices were measured in air
8.) 153
is: which suggests an effective suppression of non-irradiative recombination.
change to: which suggests an effective suppression of non radiative recombination.
9.) 244 Figure caption
Figure 5a shows the spectral photoresponse of the device in the wavelength range of 400-800 nm
Is in contradiction to line 238
(a) Spectral photoresponse performance of the S2 device in the wavelength ranging from 238
400 to 700 nm under a fixed incident light intensity

Author Response
We really thank the reviewer for reviewing this manuscript. Our modifications and clarification in response to the reviewer’s comments are listed in the following.
Comment #1: Major critic point:
Fig. 5a) shows EQE larger than 400%!!!
in line 250 an explanation is given:
The photoconduction results from the electron-hole pairs excited by the illumination light with photon energy larger than the semiconductor band gap. The 405 nm light with larger energy can excite more electron-hole pairs in perovskite thin films, leading to a higher photocurrent.
But 405 nm = 3.06 eV and the MAPbI3 energy gap is 1.6 eV to excite 2 electrons with 1photon at least 3.2 eV would be needed.
Possibly the area of the detector, which is given in line 94 as 0.014 cm2 is way to small it would be 0.12 mm x 0.12 mm.
In addition EQE is a term in photovoltaics; for detectors I find in Wikipedia:
Quantum efficiency of Image Sensors: Quantum efficiency (QE) is the fraction of photon flux that contributes to the photocurrent in a photodetector or a pixel. Quantum efficiency is one of the most important parameters used to evaluate the quality of a detector and is often called the spectral response to reflect its wavelength dependence. It is defined as the number of signal electrons created per incident photon. In some cases it can exceed 100% (i.e. when more than one electron is created per incident photon).
Spectral responsivity
Spectral responsivity is a similar measurement, but it has different units: amperes per watt (A/W); (i.e. how much current comes out of the device per incoming photon of a given energy and wavelength). Both the quantum efficiency and the responsivity are functions of the photons' wavelength (indicated by the subscript λ).
To convert from responsivity (Rλ, in A/W) to QEλ [6] (on a scale 0 to 1):
where λ is the wavelength in nm, h is the Planck constant, c is the speed of light in a vacuum, and e is the elementary charge.
Response #1: Thanks for the reviewer’s comments.
ITO substrates used in the experiment were etched to form electrodes with a channel width of 0.02 cm and length of 0.7 cm, respectively. Thus, the active area of the detector is 0.02 cm×0.7 cm (0.014 cm2). We have added this experimental detail in line 95 of the revised manuscript.
Regarding the EQE value, we have recalculated and double confirmed the results. In our calculation, the responsivity and EQE show almost the same evolution as shown in Figure 5 c and d. For photoresistors, the EQE value is also dependent on applied external bias. An external bias will result in an increasing number of electrons collected by the electrode and giving an EQE number higher than 100%. In our experiments, the EQE measurement was calculated from responsivity at 5 V bias. In addition, EQE of perovskite-based photoresistors with a value higher than 100% was also reported by other workers. For example, in Tong’s report, halide perovskite (Eg=1.7 eV) based photodetectors achieved an EQE value of 4000% at 1 V bias under 650 nm illumination with an intensity of 0.2 mW/cm2, as shown in the following Figure (Figure 4e of reference 51).
Comment #2: luminescence around PbI2 gap i.e. 500nm should be shown?
Response #2: Thanks for the reviewer’s kind suggestion. The luminescence of the perovskite films was measured using a 532 nm laser as excitation light. Therefore, the luminescence around 500 nm is not detectable. However, the present of PbI2 was confirmed by the XRD spectra, as shown in Figure 1a.
Comment #3: Red shift with increasing PbI2/MAI ratio related to reduced lattice strain: any hint on strain in XRD?
Response #3: Thanks for the reviewer’s comments. We have carefully analysed the XRD pattern again, however for the time being we cannot find direct evidence regarding the reduced lattice strain from XRD. Regarding the PL shift, lattice stain dependence have been discussed in the revised manuscript.
Line 145- 160: “In a recent work, Jones et al demonstrated that lattice strain is directly associated with enhanced defect concentrations and non-radiative recombination on the micro-sacle [44]. Here, the PL intensities are stronger for the PbI2 rich perovskites compared with the pure-phase perovskite, which are consistent with previous reports [38, 39, 45]. The presence of PbI2 can passivate the grain boundaries of the perovskite films and this passivation effect on PL intensity is dominant compared with lattice strain. The sample with MAI: PbI2 ratio of 0.5: 1 (sample S2) shows the strongest PL emission, which suggests an effective suppression of non-radiative recombination.”
Comment #4: Line 163: Considering the steady-state PL results as shown in Figure 2b, the observed longer PL lifetime in sample S2 suggests that the grain boundaries of MAPbI3
But: In Fig.2b) higher luminescence is observed, not lifetime
Response #4: Thanks for the reviewer’s comments. The defect states or shallow trapping levels in the grain boundaries of the perovskite will result in PL emission quenching and therefore behave as non-radiative recombination centers. As shown by Figure 2c, the observed longer PL lifetime in sample S2 suggests that the defect states or shallow trapping levels in the grain boundaries of the S2 sample are passivated by PbI2 in the film. Meanwhile, the steady-state PL results in Figure 2b shows a higher PL intensity was observed from the S2 sample, which is consistent with the lifetime measurement. The revision is shown in line 169-175 of the revised manuscript highlight.
Comment #5: Line 18
is: in this contribution, we prepared high …
change to: We prepared high…
Response #5: Thanks for the reviewer’s comments. The change has been made, as shown in line 18 of the revised manuscript highlight.
Comment #6: Line 28
is: the special dependence of photocurrent generation
change to: the spacial dependence of photocurrent generation
or change to: the local dependence of photocurrent generation
Response #6: Thanks for the reviewer’s comments. The change has been made, as shown in line 28 of the revised manuscript highlight.
Comment #7: Line 52
is: For a certain perovskite, composition is a crucial factor…
change to: For a certain perovskite material of AMX3 stoichiometry, chemical composition is a crucial factor…
Response #7: Thanks for the reviewer’s comments. The change has been made, as shown in line 53 of the revised manuscript highlight.
Comment #8: 102
is: photoelectronic performance of the devices were measured in the air
change to: photoelectronic performance of the devices were measured in air
Response #8: Thanks for the reviewer’s comments. The change has been made, as shown in line 103 of the revised manuscript highlight.
Comment #9: 153
is: which suggests an effective suppression of non-irradiative recombination.
change to: which suggests an effective suppression of non radiative recombination.
Response #9: Thanks for the reviewer’s comments. The change has been made, as shown in line 158 of the revised manuscript highlight.
Comment #10: 244 Figure caption
Figure 5a shows the spectral photoresponse of the device in the wavelength range of 400-800 nm
Is in contradiction to line 238
(a) Spectral photoresponse performance of the S2 device in the wavelength ranging from 238 400 to 700 nm under a fixed incident light intensity.
Response #10: Thanks for the reviewer’s comments. The change has been made, as shown in line 255 of the revised manuscript highlight.

Reviewer 2 Report
The authors report on photodetectors based on MAI: PbI2 perovskites and study the effect of different MAI-PbI2 ratio on the detected current, while highlighting the beneficial effect of PbI2 as passivating agent in the perovskite film. The manuscript is well written and suitable for publication in Nanomaterials after the following points are addressed. Although this does not affect the quality of this work, the title is misleading, the term ‘photodetectors’ should be changed into ‘photoresistors’. I got confused when I read ‘photodetectors’ as I was expecting a paper on ‘photodiodes’.
Line 38-39: mention recent progress in perovskite solar cells efficiency and perovskite on silicon tandem solar cells which have now reached 28% (cite Prof. Henry Snaith’s review https://doi.org/10.32386/scivpro.000004)
Line 39: please add more references (e.g. reviews) related to the explosion of the perovskite research field since 2012 (e.g. Snaith’s papers). Simply referring to the NREL charts is not enough. The NREL reference is wrong (add date in which the chart was retrieved).
Line 45: refer to the recent literature of peroskite LEDs (cite Prof. Richard H. Friend's review https://doi.org/10.32386/scivpro.000008).
Line 92. The MAI: PbI2 notation for the ratio is confusing, use 1:1 or similar notation, or refer to the percentage of PbI2.
Line 123 and following discussion. Sample names S1-S4 are hard to follow. Because you are focussing on the PbI2 content it might be appropriate to refer to the percentage of PbI2 in the blend rather than the MAI: PbI2 ratio, when samples are mentioned.
Line 119. Remove black bar from Figure 1. The S4 XRD line profile would be the line profile of perovskite without residual PbI2 (both from excess or degradation). However, you refer to S4 as to MAI to PbI2 ratio of 1. Again the way ratios are indicated is confusing. Probably you mean MAI: PbI2 ratio 1:1. Again please redefine ratios in a meaningful way.
Line 123-128. There is plenty of literature on XRD measurements of perovskites. The references reported in the text are not sufficient.
Table 1. This table is misplaced and should be placed around line 154. Errors in the fit should be mentioned in the table (alternatively you can plot the confidence intervals in Figure 2c, d).
Line 144. ‘As the MAPbI3 amount increases from S1 to S4’. This is confusing. I would rather write ‘as the amount of PbI2 increases in the precursor solution (or in the perovskite film)’.
Line 146-147. The sentence should be also referenced to the work of Roldán-Carmona ref. [35]. The effect of lattice strain on non-radiative losses (visible in PL measurements) in perovskites is also discussed by Jones et al. (https://doi.org/10.1039/C8EE02751J). You should add a comment on this work in relation to the observed measurements.
Line 182. Add ~ to 4.93nm. Include error in the measurement.
Figure 3. The authors should repeat the PC-AFM measurements because their quality is very poor. PC-AFM and topography AFM scans look completely different. The PC-AFM scans are extremely noisy, maybe the tip was scratching the sample. You can find examples of good PC-AFM scans in this paper: https://pubs.rsc.org/en/content/articlelanding/2016/ee/c6ee01889k#!divAbstract. Alternatively a comment on why the quality of the scans is poor should be added.
Line 189-191. I would expect to see the grain boundaries in the PC-AFM scans corresponding to the perovskite crystallites in the topography scans.
Line 208. ‘MAI: PbI2 ratio of 0.5:1 (sample S2)’. This is the correct way of reporting the MAI to PbI2 ratio.
Figure 5b. Probably the device should be called a photoresistor, but this is a minor point. When I first read the manuscript I was expecting to see a photodiode.
Author Response
We really thank the reviewer for reviewing this manuscript. Our modifications and clarification in response to the reviewer’s comments are listed in the following.
Comment #1: Line 38-39: mention recent progress in perovskite solar cells efficiency and perovskite on silicon tandem solar cells which have now reached 28% (cite Prof. Henry Snaith’s review https://doi.org/10.32386/scivpro.000004)
Response #1: Thanks for the reviewer’s comment. We have studied this review carefully and cite it as reference [5] in line 39-41 of the revised manuscript highlight.
Comment #2: Line 39: please add more references (e.g. reviews) related to the explosion of the perovskite research field since 2012 (e.g. Snaith’s papers). Simply referring to the NREL charts is not enough. The NREL reference is wrong (add date in which the chart was retrieved).
Response #2: Thanks for the reviewer’s comments. We have added more references (reference 2-4) related to the explosion of the perovskite research field since 2012 in line 39 to replace NREL reference.
Comment #3: Line 45: refer to the recent literature of peroskite LEDs (cite Prof. Richard H. Friend's review https://doi.org/10.32386/scivpro.000008).
Response #3: Thanks for the reviewer’s comment. We have cited Prof. Richard H. Friend’s review as reference 26, as shown in line 45 of the revised manuscript highlight.
Comment #4: Line 92. The MAI: PbI2 notation for the ratio is confusing, use 1:1 or similar notation, or refer to the percentage of PbI2.
Response #4: Thanks for the reviewer’s comment. We have changed the MAI: PbI2 notation to 0.25:1, 0.5:1, 0.75:1, and 1:1, as shown in line 93-94 of the revised manuscript.
Comment #5: Line 123 and following discussion. Sample names S1-S4 are hard to follow. Because you are focussing on the PbI2 content it might be appropriate to refer to the percentage of PbI2 in the blend rather than the MAI: PbI2 ratio, when samples are mentioned.
Response #5: Thanks for the reviewer’s suggestions. We have revised the sample name and rewrite the description through the manuscript. For example, “the sample with MAI: PbI2 ratio of 0.5:1 (sample S2)”.
Action taken: The revisions has been made in many places of the revised manuscript where mentioned the sample name and highlighted. All the descriptions in the revised manuscript is consistent.
Comment #6: Line 119. Remove black bar from Figure 1. The S4 XRD line profile would be the line profile of perovskite without residual PbI2 (both from excess or degradation). However, you refer to S4 as to MAI to PbI2 ratio of 1. Again the way ratios are indicated is confusing. Probably you mean MAI: PbI2 ratio 1:1. Again please redefine ratios in a meaningful way.
Response #6: Thanks for the reviewer’s constructive suggestion. We have removed the black bar from Figure 1 and the ratios have also been redefined.
Comment #7: Line 123-128. There is plenty of literature on XRD measurements of perovskites. The references reported in the text are not sufficient.
Response #7: Thanks for the reviewer’s comment. We have added more references on XRD measurements of perovskites, as shown in line 129 of the revised manuscript highlight.
Comment #8: Table 1. This table is misplaced and should be placed around line 154. Errors in the fit should be mentioned in the table (alternatively you can plot the confidence intervals in Figure 2c, d).
Response #8: Thanks for the reviewer’s good suggestion. We have moved table 1 to the place assigned by the reviewer and errors in the fit have been added in the table, as shown in line 159.
Comment #9: Line 144. ‘As the MAPbI3 amount increases from S1 to S4’. This is confusing. I would rather write ‘as the amount of PbI2 increases in the precursor solution (or in the perovskite film)’.
Response #9: Thanks for the reviewer’s comment. We have changed the sentence to “As the MAI: PbI2 ratio varies from 0.25:1 to 1:1”, as shown in line 145-146 of the revised manuscript.
Comment #10: Line 146-147. The sentence should be also referenced to the work of Roldán-Carmona ref. [35]. The effect of lattice strain on non-radiative losses (visible in PL measurements) in perovskites is also discussed by Jones et al. (https://doi.org/10.1039/C8EE02751J). You should add a comment on this work in relation to the observed measurements.
Response #10: Thanks for the reviewer’s comment. We have carefully studied this report and cite it as reference 44 in the revised manuscript. This paper helps us explain the effect of the lattice strain more clearly.
Action taken: The following revision has been made on Page 5 Line 151-156:
“In a recent work, Jones et al observed that lattice strain is directly related to enhanced defect concentrations and non-radiative recombination on the micro-scale [44]. Here, the PL intensities are stronger for the PbI2 rich perovskites compared with the pure-phase perovskite, which are consistent with previous reports [34, 35, 40]. The presence of PbI2 can passivate the grain boundaries of the perovskite films and this passivation effect on PL intensity is dominant compared with lattice strain.”
Comment #11: Line 182. Add ~ to 4.93nm. Include error in the measurement.
Response #11: Thanks for the reviewer’s comment. We have added “~”to 4.93 nm in line 193 of the revised manuscript.
Comment #12: Figure 3. The authors should repeat the PC-AFM measurements because their quality is very poor. PC-AFM and topography AFM scans look completely different. The PC-AFM scans are extremely noisy, maybe the tip was scratching the sample. You can find examples of good PC-AFM scans in this paper: https://pubs.rsc.org/en/content/articlelanding/2016/ee/c6ee01889k#!divAbstract. Alternatively a comment on why the quality of the scans is poor should be added.
Response #12: Thanks for the reviewer’s comment. We repeated the PC-AFM measurements and tried our best to improve its quality in order to reduce scan noise and observe grain boundaries. We used the “peakforce tapping” mode provided by Bruker Dimension Icon atomic force microscope. With carefully tuning of the scan conditions (gain, Z limit, scan rate), the scan noise is reduced and the image quality has been improved, as shown in following figure.
Comment #13: Line 189-191. I would expect to see the grain boundaries in the PC-AFM scans corresponding to the perovskite crystallites in the topography scans.
Response #13: Thanks for the reviewer’s comment. We repeated the PC-AFM measurement and the grain boundaries can be observed, which is corresponding to the topography scans.
Comment #14: Line 208. ‘MAI: PbI2 ratio of 0.5:1 (sample S2)’. This is the correct way of reporting the MAI to PbI2 ratio.
Response #14: Thanks for the reviewer’s comment. We have revised the way of reporting the MAI to PbI2 ratio according to the suggestion.
Comment #15: Figure 5b. Probably the device should be called a photoresistor, but this is a minor point. When I first read the manuscript I was expecting to see a photodiode.
Response #15: Thanks for the reviewer’s suggestion. We agree that a photoresistor is more accurate to describe the devices and we have changed the name to photoresistors.

Round 2
Reviewer 1 Report
the authorth responded in an adequate way to my critique points
Author Response
We really thank the reviewer for reviewing this manuscript.
Reviewer 2 Report
I still see some minor mistakes. Please correct them. The rest of the manuscript is fine.
Reference [5] is wrong, the correct reference is ‘H. J. Snaith & S. Lilliu. The Path to Perovskite on Silicon PV. Scientific Video Protocols 1, 1, (2018)’.
Reference [26] is wrong, the correct reference is ‘R. H. Friend, D. Di, S. Lilliu & B. Zhao. Perovskite LEDs. Scientific Video Protocols 1, 1, (2019)’
Line 72: Change ‘photoresistor-style’ into just ‘photoresistor’.
Line 155. Change ‘et al’ into ‘at al.’
Author Response
We really thank the reviewer for reviewing this manuscript. Our modifications and clarification in response to the reviewer’s comments are listed in the following.
Comment #1: Reference [5] is wrong, the correct reference is ‘H. J. Snaith & S. Lilliu. The Path to Perovskite on Silicon PV. Scientific Video Protocols 1, 1, (2018)’.
Response #1: Thanks for the reviewer’s comment. We have corrected reference [5] in the right way, as shown in line 327 of the revised manuscript highlight.
Comment #2: Reference [26] is wrong, the correct reference is ‘R. H. Friend, D. Di, S. Lilliu & B. Zhao. Perovskite LEDs. Scientific Video Protocols 1, 1, (2019)’.
Response #2: Thanks for the reviewer’s comments. We have corrected reference [26] in the right way, as shown in line 376 of the revised manuscript highlight.
Comment #3: Line 72: Change ‘photoresistor-style’ into just ‘photoresistor’.
Response #3: Thanks for the reviewer’s comment. We have changed ‘photoresistor-style’ into ‘photoresistor’, as shown in line 72 of the revised manuscript highlight.
Comment #4: Line 155. Change ‘et al’ into ‘et al.’
Response #4: Thanks for the reviewer’s comment. We have changed ‘et al’ into ‘et al.’, as shown in line 155 of the revised manuscript.

This manuscript is a resubmission of an earlier submission. The following is a list of the peer review reports and author responses from that submission.
Round 1
Reviewer 1 Report
This manuscript demonstrates high-efficiency perovskite photodetectors using PbI2 doping method. The authors systematically investigated the effect of the doping level on the structure and photodiode performance of the perovskite layer. And, the authors found the passivation effect of PbI2 and showed the improved detector performance. So, it is recommended to accept the manuscript but with some minor revisions:
1. In label of Table 1, tau symbols are missing.
2. The author explained that PbI2 can passivate the grain boundary of the perovskite grain. However, PbI2-rich perovskite layer has smaller grain size compared to pure perovskite layer. Is it possible to passivate the pure perovskite layer by additionally coating the PbI2 solution on the pure perovskite layer, like the post-coating layer?
3. As shown in Figure 3d, the photocurrent is higher at the grain boundary. Even though the grain boundary is passivated by PbI2, but still grain boundary where the recombination rate is higher than that in the bulk. Why is the photocurrent at the grain boundary is higher than that at the center of single-crystalline perovskite grain?
4. In page 6 line 200-202, the two sentences should be connected by “,”, not “.”.
5. Why is the dark current of Figure 5b is two orders lower than that of Figure 4b at ±4V?
Author Response
We really thank the reviewer for reviewing this manuscript. Our modifications and clarification in response to the reviewer’s comments are listed in the following.
Point 1: In label of Table 1, tau symbols are missing.
Response #1: Thanks for the reviewer’s comments. Tau symbols have been added in Table 1.
Point 2: The author explained that PbI2 can passivate the grain boundary of the perovskite grain. However, PbI2-rich perovskite layer has smaller grain size compared to pure perovskite layer. Is it possible to passivate the pure perovskite layer by additionally coating the PbI2 solution on the pure perovskite layer, like the post-coating layer?
Response #2: Thanks for the reviewer’s kind suggestion. We agree that a post-coating layer of PbI2 on the pure perovskite layer may show some passivation effects. However, the passivation effects may not be as good as the homogeneously mixed PbI2 and perovskite. Since the PbI2 would only contact the surface of the pure perovskite and the passivation effects could be limited. Meanwhile, pure perovskites are shown to be sensitive to heat and strong light illumination, and the formation of a homogeneously mixed PbI2/Perovskite layer could also be more effective to prevent the perovskite decomposition, and enhance the stability of perovskite.
Point 3: As shown in Figure 3d, the photocurrent is higher at the grain boundary. Even though the grain boundary is passivated by PbI2, but still grain boundary where the recombination rate is higher than that in the bulk. Why is the photocurrent at the grain boundary is higher than that at the center of single-crystalline perovskite grain?
Response #3: Thanks for the reviewer’s comments. Compared with the centre of single-crystalline perovskite grain, carriers are more likely to transfer along the grain boundaries or surfaces in polycrystalline perovskite film. Our results show that when the grain boundary is passivated by PbI2, the defects at grain boundaries can be reduced which results in more photogenerated charge carriers transferring along the grain boundaries. Therefore, the photocurrent at the grain boundary becomes further enhanced compared with the centre of single-crystalline perovskite grain.
Action taken: To make the discussion clearer, we have revised the corresponding sentences
on page 6 Line 190-196:
“…, which could be due to the grain boundaries passivation effect of PbI2. When the grain boundary is passivated by PbI2, the defects at grain boundaries can be reduced, resulting in more photogenerated charge carriers transferring along the grain boundaries. Therefore, the photocurrent at the grain boundary becomes further enhanced compared with the center of single-crystalline perovskite grain (as shown in Figure 3d). In addition, the current photoresponse at 1 V is extracted as shown in Figure 3e, which clearly demonstrates a higher current density in the PbI2 rich perovskite thin film.”
Point 4: In page 6 line 200-202, the two sentences should be connected by “,”, not “.”.
Response #4: Thanks for the reviewer’s comments. In the revised manuscript, we replaced “.” by “,”.
Point 5: Why is the dark current of Figure 5b is two orders lower than that of Figure 4b at ±4V?
Response #5: Thanks for the reviewer’s comments. In Figure 4b, we showed the photocurrent of different devices under light illumination and the corresponding dark currents of these devices are shown in Table 2, which are close to the dark current of Figure 5b (~10-8A or ~10-2μA).

Reviewer 2 Report
Authors compared photodetectors based on MAPbI3 film with different PbI2 doping levels. Unless the following issues can be answered appropriately first, I don’t suggest to publish this paper.
1.Pb is the environmentally unfriendly substance. In this study, the authors fixed the amount of PbI2 at 1 mol/L, and change the MAI amount. Their conclusion shows that the low MAI amount (PbI2 rich) results in better device performance. This result doesn’t seem to be good for environment.
2.Authors claimed that saturation is observed in many studied, such as Ref. 46. However, curves in Ref. 46 shows saturation at ~ 20 mW/cm2, which is much larger than authors’ results with saturation at ~0.5 mW/cm2. I am afraid that the accuracy of results measured by authors’ power meter. For example, TES-132 is used for power measurement for high intensity of solar irradiance of ~ 100 mW/cm2.
Author Response
We really thank the reviewer for reviewing this manuscript. Our modifications and clarification in response to the reviewer’s comments are listed in the following.
Point 1: Pb is the environmentally unfriendly substance. In this study, the authors fixed the amount of PbI2 at 1 mol/L, and change the MAI amount. Their conclusion shows that the low MAI amount (PbI2 rich) results in better device performance. This result doesn’t seem to be good for environment.
Response #1: Thanks for the reviewer’s comment. The reviewer raised an important point for the current perovskite studies. We agree that environmental hazards are one of the most important concerns for perovskite-based devices. However, exploring lead-free perovskites and improving their performance close to lead containing perovskites is still a big challenge. In this contribution, we show that the PbI2 rich perovskite possesses better performance and stability, which may still be interesting for researchers in this field, and may even stimulate relevant studies on the lead-free perovskite optoelectronic devices. Briefly speaking, we have shown the basic design principles of perovskite-based optoelectronic devices in this study. Based on the PCAFM, XRD, electrical measurements and etc, those results showed that the non-stoichiometric content of organic and inorganic composition would be beneficial to the device performance and stability, due to the scientifically enhanced photogenerated charge carriers in such heterojunction systems.
Actions Taken: We have added corresponding discussion on the development of environmentally friendly perovskite-based optoelectronic devices:
On Page 9, Line 291-294: “These results indicate that non-stoichiometric perovskites could be beneficial to the device performance and chemical stability, which could also be extended to the exploration of lead-free perovskites for the high-performance photodetector application.”
Point 2: Authors claimed that saturation is observed in many studied, such as Ref. 46. However, curves in Ref. 46 shows saturation at ~ 20 mW/cm2, which is much larger than authors’ results with saturation at ~0.5 mW/cm2. I am afraid that the accuracy of results measured by authors’ power meter. For example, TES-132 is used for power measurement for high intensity of solar irradiance of ~ 100 mW/cm2.
Response #2: Thanks for the reviewer’s comment. We have carefully studied the Ref. 46. (High-Performance Red-Light Photodetector Based on Lead-Free Bismuth Halide Perovskite Film. ACS Appl. Mater. Interfaces 2017, 9, 18977-18985). Ref. 46 mainly focuses on the photoresponse studies of lead-free bismuth halide perovskite thin film, which is a quite different material type (CsBi3I10) compared with the MAPbI3 studied in our work. In fact, the power dependent studies of perovskite-based photodetectors can be significantly affected by several factors including wavelength of incident light, material thickness, device structure, metal contact, bias voltage applied, etc. In Ref. 46, the saturation at a higher light intensity was observed with 650 nm light. However, our results with saturation at ~0.5 mW/cm2 was obtained under 405 nm light, which is in agreement with the previous report (Ref. 8, Adv. Mater. 2016, 28, 3683–3689, Fig3 a, b), where the saturation is gradually reached when the light intensity increased from 0.1 to 7mW cm−2.
The light intensity of monochromatic light (405 nm) was measured by a Thorlabs power meter (PM100D with a S121C standard photodiode power sensor). The S121C sensor head has a wavelength range from 400 nm to 1100nm, and power range from 500 nW to 500 mW with a resolution of 10 nW. Meanwhile, light intensity of white light (as shown in Figure 4) were estimated by TES-132, which has a resolution of 0.1W/m2.
Action taken: To make the experimental details clearer, the following revision has been made on Page 3 Line 106-108:
“The light intensities of white light and monochromatic light were recorded by TES-132 solar power meter and a Thorlabs power meter (PM100D with a S121C standard photodiode power sensor), respectively.”

Round 2
Reviewer 2 Report
The authors reply can't convinced me. Since authors also agreed that lead-free perovskites is desiring, the claimed PbI2 rich process didn't interest researchers especially for readers of this high-impact Journal. Authors mentioned that their results agreed with those in Ref. 8. However, Fig. 3c of Ref. 8 is plotted in log scale, and redrawing of Fig. 3c of Ref. 8 in linear scale shows the saturation occurs after 4 mW/cm2, which is also much larger than authors’ results with saturation at ~0.5 mW/cm2